# Attribution of global evapotranspiration trends based on the Budyko framework

Shijie Li[a], Guojie Wang[a], Chenxia Zhu[a], Jiao Lu[a], Waheed Ullah[a], Daniel Fiifi Tawia Hagan[a], Giri Kattel[a,b,c], Jian Peng[d,e]

[a] Collaborative Innovation Center on Forecast and Evaluation of Meteorological Disasters (CIC–FEMD), School of Geographical Sciences, Nanjing University of Information Science and Technology, Nanjing 210044, China
[b] Water and Agriculture Program (WEAP), Department of Infrastructure Engineering, The University of Melbourne, Melbourne 3010, Australia
[c] Department of Hydraulic Engineering, Tsinghua University, Beijing 100084, China
[d] Department of Remote Sensing, Helmholtz Centre for Environmental Research-UFZ, Permoserstrasse 15, 04318, Leipzig, Germany
[e] Remote Sensing Centre for Earth System Research, Leipzig University, Talstr. 35, 04103, Leipzig, Germany

*Correspondence to*: Guojie Wang (gwang@nuist.edu.cn)

**Abstract.** Actual evapotranspiration (ET) is an essential variable in the hydrological process, linking the carbon, water, and energy cycles. Global ET has significantly changed in the warming climate. Although increasing vapour pressure deficit (VPD) due to global warming enhances atmospheric water demand, it remains unclear how the dynamics of ET are affected. In this study, using multiple datasets, we disentangled the relative contributions of precipitation, net radiation, air temperature ($T_1$), VPD, and wind speed on affecting annual ET linear trend using an advanced separation method that considers the Budyko framework. We found that the precipitation variability dominantly controls global ET in the dry climates while the net radiation has substantial control over ET in the tropical regions, and VPD impacts ET trends in boreal mid-latitude climate. The critical role of VPD in controlling ET trends is particularly emphasized due to its influence in controlling carbon-water-energy cycle.

## 1 Introduction

Actual evapotranspiration (ET) is when water transforms from a liquid to a gaseous state. Such transformation synchronously absorbs the air's energy, making ET the largest terrestrial water flux component accounting for >60% of global land precipitation (Trenberth et al., 2007). ET directly affects hydrological processes at regional and global scales (Zhang et al., 2016) by linking water, energy, and carbon cycles. As a result, ET plays a crucial role in land-atmosphere interactions amongst various climatic variables, including precipitation, air temperature, humidity, solar radiation and wind speed (Koster et al., 2006; Wang et al., 2011; Miralles et al., 2018), which consequently influence the climate at regional and global scales. An accurately estimated ET can therefore provide a comprehensive contribution to understanding the changes in hydrological cycles and the associated extreme events, such as droughts and floods, as well as their impacts on ecosystem productivity, water-use efficiency, and irrigation (Sheffield et al., 2012; Sun et al., 2017; Jalilvand et al., 2019).

However, the available ground ET measurements from traditional methods (e.g. eddy covariance, porometry and lysimeters, and scintillometry) have shortcomings such as sparse observational sites and short time span (Allen et al., 1991; Everson et al., 2009; Monteith et al., 1990). To overcome these limitations, various spatially distributed ET products have been developed and widely used, including those from remote sensing, land surface models, and reanalysis, such as the GLEAM (Global Land Evaporation Amsterdam Model) product and the GLDAS (Global Land Data Assimilation System) (Miralles et al., 2011a, b; Mu et al., 2011; Reichle et al., 2017; Loew et al., 2016; Peng et al., 2020).

Studies of climate warming intensification on global water cycle have indicated the increasing importance of understanding ET changes in space and time (e.g., Allen et al., 2002; Wu et al., 2013; Pan et al., 2015). In recent decades, ET has shown sudden increasing or decreasing trends across the globe (Miralles et al., 2013; Pascolini-Campbell et al., 2019), so an accurate attribution of the ET changes is urgently needed. The changing ET over the longer time scale is jointly determined

by climatic modes (e.g. El Niño-Southern Oscillation) (Martens et al., 2018; Miralles et al., 2013) and the long-term changes of climatic variables (Pan et al., 2020; Zeng et al., 2016; Rigden et al., 2016). For example, the upward trend of the global ET from 1982 to the late 1990 was attributed to increased radiation and air temperature (Douville et al., 2013; Jung et al., 2010). Similarly, the change of global ET during 1998-2008 lapsed due to limited soil moisture supply in the Southern Hemisphere and transitions to the El Niño condition (Jung et al., 2010; Miralles et al., 2014). Zhang et al. (2015) demonstrated how the water supply, available energy, and atmospheric water demand jointly affected the global ET changes from 1982 to 2013, accounting for 49%, 32%, and 19% of global ET changes, respectively.

However, different evapotranspiration algorithms, parameterizations, and input climate forcing datasets can cause uncertainties when attributing global ET changes (Vinukollu et al., 2011; Michel et al., 2016). For example, Miralles et al (2016) evaluated the performances of three models using the same forcing data, finding that the GLEAM product was relatively better than the other global ET products over most wet and dry conditions. Similarly, Badgley et al (2015) used 19 different combinations of input forcing datasets to run the Priestly-Taylor Jet Propulsion Laboratory (PT-JPL) ET model, indicating that the choice of forcing datasets accounted for an average 20% error. Those results have indicated that the improper choice of ET models and forcing data may add significant uncertainties to the ET attributions.

Potential ET (PET) is determined by radiation, air temperature, VPD, and wind speed, and it reflects the magnitude of atmospheric demand on land ET. Against the backdrop of a warming climate, rising air temperature has increased atmospheric water demand (i.e. increased PET) (Fu et al., 2014; Feng et al., 2013; Dai et al., 2004). Studies have indicated that increased VPD primarily determines the recent PET increase, a function of air temperature and humidity (Dai et al., 2017; Ficklin et al., 2017). Increased VPD tends to make plants close their stomata to avoid water loss and thus restrain transpiration (Novick et al., 2016; McAdam et al., 2015). A high atmospheric water demand induced by VPD promotes PET, while the increased surface resistance limits ET. Li et al. (2021) have found VPD has dominated the increase of annual ET in energy-limited regions such as southeastern China. However, it's not clear how VPD affects global long-term ET changes.

Furthermore, the above studies focused on the influences of climatic variables on long term ET changes, however, a distinct shortcoming in these studies is that they only demonstrated the responses of long term ET change (variance) to certain factors (e.g. climatic variables and surface conductance). A few studies disentangle the contributions of relatively complete climatic factors (mainly atmospheric), including precipitation, net radiation, air temperature, VPD, and wind speed, to the annual ET linear trend. Along these lines, Li et al (2021) attempted to quantify the contribution of those forcing variables to ET trends over China with the Budyko theory. Their study indicates that precipitation dominates ET trends over water-limited regions, while VPD controls ET of energy-limited regions. However, there are still unclear questions about the global land ET mechanism. For example, how differently would the conclusions of dominating ET factors over water-limited regions be for global dry lands? Which variable controls ET over the global tropical zone is unclear, despite the results of VPD controlling ET over the energy-limited region of China. Wang et al (2022) indicate that global significantly increased ET mostly results from increasing air temperature, especially in the humid region. Pan et al (2020) point out that precipitation, air temperature, and radiation control Amazon's ET changes. On the other hand, the boreal ET mechanisms are also not entirely clear. Increasing air temperature is significantly correlated with ET (Wang et al., 2022), while increasing VPD contributes to ET process over the boreal region (Helbig et al., 2020). Therefore, it is necessary to assess global ET mechanisms using the same attribution method for solving these problems.

In this study, we have adopted the Budyko theory to advance our understanding of the response of global ET trends to climatic variables, including precipitation (P), net radiation ($R_n$), air temperature ($T_1$), VPD, and wind speed (u). The Budyko theory investigates the interactions between ET, PET, and P (Yokoo et al., 2008; Yang et al., 2008; Liu et al., 2011). For example, Teuling et al. (2019) explored the dynamics of ET in Europe at high resolution (1 km$^2$) with the Budyko model and key meteorological variables. Here we use multiple datasets such as GLEAM3.0a, EartH2Observe ensemble (EartH2Observe-En), GLDAS2.0-Noah, and MERRA-Land (Modern Era Retrospective-Analysis for Research and

Application-Land). Using multiple datasets can reduce uncertainties of the forcing data to accurately attribute global ET changes over different land covers and climate regimes (Li et al., 2018).

## 2 Data and Methods

### 2.1 Data

We use multiple ET products and their respective forcing data, including the remote sensing-based GLEAM product, land surface model ensembled product (EartH2Oberve-En), and two reanalysis products (GLDAS2.0-Noah and MERRA-Land). These products have different temporal lengths, and we have used their overlapping period from 1980 to 2010. In the attribution method with Budyko framework, we use respective forcing data of each product (please see detail description in section 2.2 Forcing data). To study the ET mechanism within different climatic conditions, we have divided the global land into Tropical, Dry, Mild Temperate, Snow, and Polar zones, respectively, using the Köppen climate classification (Kottek et al. 2006) (Figure 1). The Köppen climate classification is produced according to the empirical relationship between climatic variables and vegetation.

### 2.1.1 ET products

a) GLEAM3.0a ET

The GLEAM3.0a is arguably the longest of various ET products, mainly determined from remote sensing observations. It consists of soil evaporation, canopy transpiration, interception loss, snow sublimation, and open-water evaporation. A key feature of this product is the use of the Gash analytical model to estimate interception loss. The other components in this product are calculated according to the Priestley-Taylor equation (Miralles et al., 2011b; Martens et al., 2017).

b) EartH2Oberve-En ET

The EartH2Oberve product uses ten models, including five hydrological models, four land surface models, and a simple water balance model, which are forced by the same state-of-the-art meteorological reanalysis (Dutra et al., 2015). Schellekens et al. (2017) indicated that the model ensemble outperforms the individual model's output, and thus we have used the ensembled mean data from the ten models here (regarded as EartH2Oberve-En). The EartH2Oberve-En product is demonstrated to be an accurate reanalysis data and has been used for multiple-scale water resource evaluation (Schellekens et al., 2017).

c) GLDAS2.0-Noah ET

The GLDAS was initially developed by the National Aeronautics and Space Administration (NASA) Goddard Space Flight Center (GSFC) of America, based on the North American Land Data Assimilation System (NLDAS). GLDAS is a global, high-resolution, offline terrestrial modeling system and produces the outputs of land surface states and fluxes in near-real time, such as ET, soil moisture, latent, sensible, and ground heat flux. Satellite and ground-based observations are used to constrain the forcing and parameterization of used land surface models (i.e. Mosaic, Noah, the Community Land Model, and the Variable Infiltration Capacity model) (Rodell et al., 2004). Here, the ET product derived from GLDAS2.0-Noah is used in our study.

d) MERRA-Land ET

The MERRA reanalysis is developed by NASA's Global Modeling and Assimilation Office (GMAO). It was produced by the Goddard Earth Observing System model Version 5 (GEOS-5) along with its associated data assimilation system (DAS)

Version 5.2.0 (Rienecker at al., 2011). Since there are significant errors in values and timing of precipitation in the original MERRA product (Reichle et al., 2011), we have used the offline MERRA-Land product forced by corrected precipitation. Studies have shown that the hydrological performance in the MERRA-Land has been significantly improved than in the original MERRA product (Reichle et al., 2011).

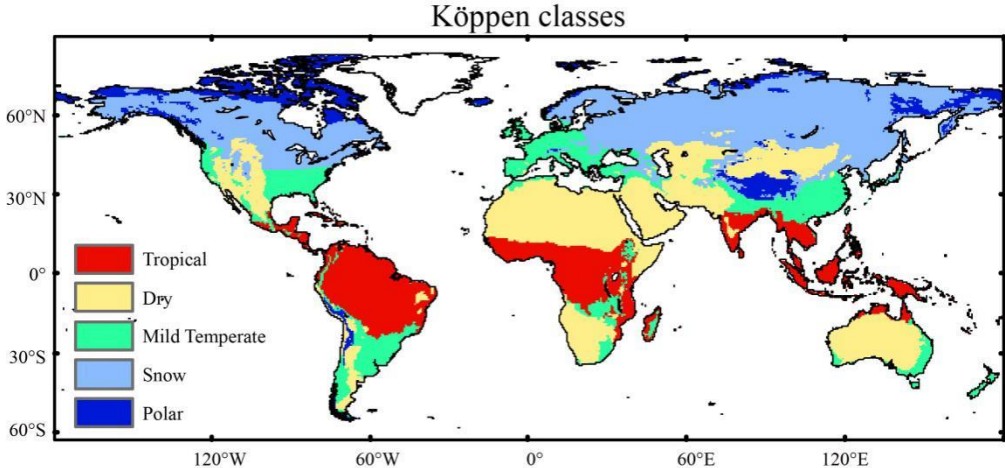

**Figure 1.** Spatial distribution of five climatic zones with Köppen climate classification, including Tropical, Dry, Mild Temperate, Snow, and Polar zone (Kottek et al., 2006).

## 2.2 Atmospheric Forcing datasets

The atmospheric forcing data in four ET products mainly include precipitation, net radiation, air temperature, specific humidity, and wind speed. The forcing data of respective ET product and their references are listed in Table 1 (Li et al., 2021). To reduce the uncertainties associated with inconsistent forcing data sources, the trends of each ET product are attributed using its own forcing data. The GLEAM algorithm does not use specific humidity and wind speed as inputs unlike the other three products. Since the other forcing data of GLEAM product (e.g. radiation and air temperature) is derived from the ERA-Interim reanalysis, specific humidity and wind speed from the same reanalysis is used for its attribution.

**Table 1.** The main forcing data in four ET products

| ET Products | Precipitation | Radiation | Air temperature | Specific humidity | Wind speed | References |
|---|---|---|---|---|---|---|
| GLEAM3.0a | MSWEP | ERA-Interim | ERA-Interim | ERA-Interim | ERA-Interim | *Martens et al., 2017* |
| EartH2Observe-En | WFDEI | WFDEI | WFDEI | WFDEI | WFDEI | *Schellekens et al., 2017* |
| GLDAS2.0-Noah | PUMFD | PUMFD | PUMFD | PUMFD | PUMFD | *Rodell et al., 2004* |
| MERRA-Land | CPC-U | MERRA | MERRA | MERRA | MERRA | *Reichl et al., 2011* |

Note: MSWEP indicates the Multi-Source Weighted-Ensemble Precipitation product; WFDEI is the Water and Global Change FP7 project forcing dataset ERA-Interim (Weedon et al., 2015; Dee et al., 2011); PUMFD indicates the meteorological forcing data of Princeton University (Sheffield et al., 2006); CPC-U is Climate Prediction Center Unified.

## 2.2 Method

### 2.2.1 Determining trends

We have used Theil-Sen's slope method to determine the trends of annual ET and climatic variables during 1980-2010. This method is nonparametric and can provide more accurate trend estimation for skewed data when compared to the linear regression approach (Wilcox, 2010). To detect the significance level of these data, we used the nonparametric Mann-Kendall test to determine the significance level of the linear trends (Mann, 1945 and Kendall, 1975). Both methods have been widely used in climate change studies (Su et al., 2015; Wang et al., 2018a; Shan et al., 2015; Shi et al., 2016).

### 2.2.2 Attribution method

The attribution method consists of two steps: firstly, building the relationship between ET and the abovementioned five climatic variables with Budyko and modified FAO Penman-Monteith equations; secondly, conducting sensitivity experiment analysis to quantify the contribution of each climatic variable to the long-term ET trends of 1980-2010.

a) Budyko relationship

The Budyko equation (Eq(1)) is usually regarded as a common way to study how climatic factors influence the annual ET changes, based on the mathematical relationships between precipitation, PET, and ET (Yokoo et al., 2008; Yang et al., 2009; Liu et al., 2011).

$$\frac{ET}{P} = 1 + \frac{PET}{P} - \left[1 + (\frac{PET}{P})^{\omega}\right]^{\frac{1}{\omega}} \quad (1)$$

where ET, PET, and P reflect ET, potential ET, and precipitation, respectively; $\omega$ indicates the landscape properties, such as vegetation cover, soil, and topography. For a particular product, pixel-wise $\omega$ can be fitted by using the least-square regression method because other variables during 1980-2010 are known (i.e. ET, P, and PET). PET reflects the atmospheric water demand, which can be determined by solar radiation, air temperature, actual vapor pressure, and wind speed etc.

There are various methods for calculating PET, including Penman-Monteith (Allen et al., 1998), Hargreaves (Hargreaves and Samani, 1985), and Priestly-Taylor methods (Priestley and Taylor, 1972). Generally, the modified FAO Penman-Monteith equation is a universal method for estimating PET with meteorological data (Allen et al., 1998). In this study, annual PET is obtained with the modified Penman-Monteith equation:

$$PET = \frac{0.408\Delta(R_n - G) + \gamma\frac{900}{T_1 + 273}uVPD}{\Delta + \gamma(1 + 0.34u)} \quad (2)$$

where $R_n$ represents net radiation, calculated by net incoming shortwave minus net outgoing longwave radiation (unit: MJ/m$^2$/yr); G is the soil heat flux density (unit: MJ/m$^2$/yr), and can be neglected on monthly or longer time scales; $\gamma$ and $\Delta$ reflect the psychometric constant and slope of the vapor pressure curve, respectively (unit: kPa/°C); $T_1$ indicates average 2-meter air temperature (unit:°C), and is used to calculate $\Delta$; $u$ indicates 2-meter wind speed (unit: m/s); VPD (kPa) is the saturation vapor pressure deficit, as a function of air temperature $T_2$ and specific humidity (i.e. VPD=f($T_2$, specific humidity)). In Eq(2), the effect of air temperature $T$ on PET is separated into two parts: $T_1$ and $T_2$. $T_1$ reflects the effect of air density and slope of the vapor pressure curve, and $T_2$ indicates the partial effect of VPD. For further details on the calculation of VPD and the difference between $T_1$ and $T_2$, refer to Allen et al. (1998).

By putting Eq(2) into Eq(1), we can obtain the direct relationship between ET and P, $R_n$, $T_1$, VPD, and $u$ as indicated in Eq. 3:

$$ET = P + \frac{0.408\Delta(R_n - G) + \gamma\frac{900}{T_1 + 273}uVPD}{\Delta + \gamma(1 + 0.34u)} - P\left[1 + (\frac{0.408\Delta(R_n - G) + \gamma\frac{900}{T_1 + 273}uVPD}{P(\Delta + \gamma(1 + 0.34u))})^{\omega}\right]^{\frac{1}{\omega}} \quad (3)$$

b) Attribution experiments

The trends of annual ET during 1980-2010 are determined by compound influences of the main climatic factors (i.e. P, $R_n$, $T_1$, VPD, and $u$). To disentangle the impact of each climatic factor, six experiments have been designed based on Eq (3), including one control experiment (sim_CTL) and five individual factor sensitivity experiments (sim_P, sim_$R_n$, sim_$T_1$, sim_VPD, and sim_$u$ respectively). The sim_CTL experiment provides the control ET changes for each product by using all the factors of 1980-2010, while the ET change controlled by a particular factor is simulated by the sensitivity experiment with the factor only in the 1980 and the others factors between 1980 and 2010. The multiyear average can also replace a factor in 1980 during 1980-2010. Figure S1 shows that precipitation and PET values between 1980 and the multiyear average are very close. For example, the ET change of 1980-2010 impacted by P (i.e. sim_P) can be computed by using the constant value of P in 1980 and $R_n$, $T_1$, VPD, and $u$ of 1980-2010. The contributions of the other climatic factors (sim_$R_n$,

sim_$T_1$, sim_VPD, and sim_$u$) can be determined similarly for each product. The contribution of each factor to ET change in each product is obtained (Sun et al., 2016 and 2017):

$$C_i = \frac{\sum_{k \neq i}^{n} E_{sim\_k} - (n-2) \cdot E_{sim\_i}}{n-1}. \qquad (4)$$

where $n$ shows the number of sensitivity experiments, and n is equal to 5 here; and $E_{sim\_i}$ indicates the $i^{th}$ sensitivity experiment. We should note that the total contribution of air temperature $T$ to ET changes here is separated into two sensitivity experiments: sim_$T_1$ and sim_VPD. The sim_$T_1$ means the effect of air temperature $T_1$ in Eq (2) on ET; the sim_VPD, controlled by air temperature $T_2$ and specific humidity, contains the contribution of air temperature $T_2$.

## 3 Results

### 3.1 Trends of the ET products

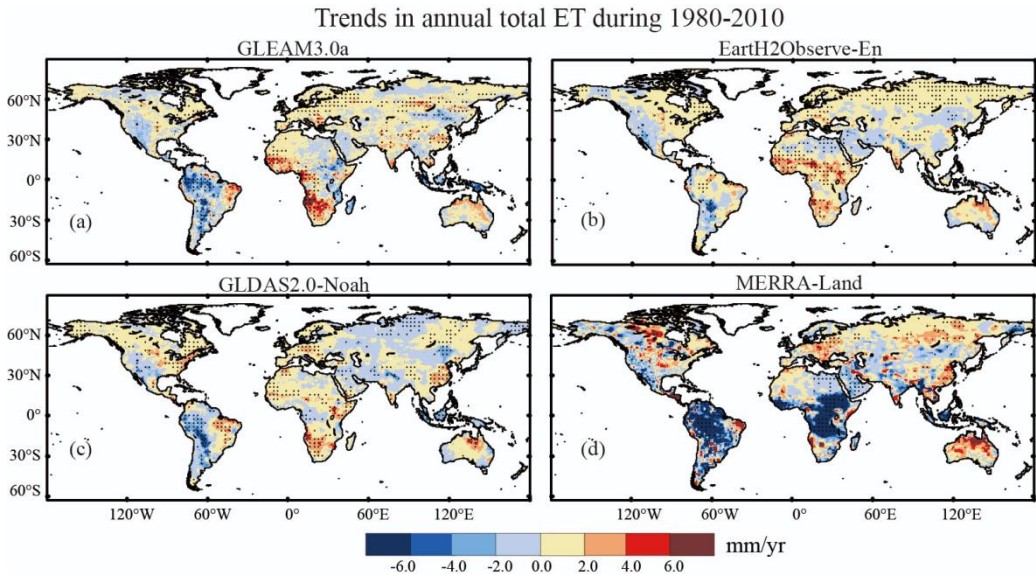

**Figure 2.** The spatial distribution of pixel-wise linear trends of annual ET for (a) GLEAM3.0a, (b) EartH2Observe-En, (c) GLDAS2.0-Noah, and (d) MERRA-Land products during 1980-2010. The trend is estimated with Theil-Sen's slope method, and the significance level is tested with the Mann-Kendall method. The dotted area indicates the trend has passed the significance test at 5% level.

The spatial distribution of long-term trends in annual ET is depicted in Figure 2, with evident differences and similarities among the four selected products in different regions. Compared to the other products, the MERRA-Land shows more significant ET changes, for example, the declining trend with a rate of about -6.0 mm/yr in Africa and South America. In most regions of the Eurasian continent, the ET changes for all products mainly amount to -2.0-2.0 mm/yr. A significant increase in ET is observed in western Europe, southeastern China, and northern Australia. In contrast, a declining ET trend is observed in northeast China and Arabian Peninsula, despite differences among the used ET products. These products show quite different trends in Africa; while the MERRA-Land indicates significantly declining ET, the trends in the other products are opposite.

### 3.2 Attributions of ET trends

The influence of each driving factor on the long-term annual ET linear trend is quantified by the attribution method in Sect. 2.2.2. Figure 3 shows the trends in each variable and its respective contribution to ET changes across different climate zones. Precipitation (P) appears to make the largest contribution, air temperature ($T_1$) and wind speed ($u$) make the smallest (except $u$ in MERRA-Land) contribution, and moderate contributions are evident for net radiation ($R_n$), and VPD. Compared to other products, a sharp decreasing $u$ in MERRA-Land leads to a decreasing ET trend. The grid number of P in

the first and third quadrants is more than 85% of the sum in Figure 3a, indicating P is positively correlated with ET. ET in the Dry zones is more sensitive to changes in P, while in Tropical zones, the effect of P is not obvious. Such a relationship also exists in $R_n$ (Figure 3b), VPD (Figure 3d), and $u$ (Figure 3e). Different contributions among climatic zones are also observed for $R_n$ (Figure 3b) and VPD (Figure 3d). For example, the $R_n$ contribution in the Tropical zone exceeds that in the Mild Temperate zone, while the VPD contribution in the Mild Temperate zone is larger than that in the Tropical zone. Limited grids, mostly from the water-limited region (Dry), fall in the fourth quadrant, amounting to 25.26%-41.96% of the sum, suggesting increasing $T_1$ hinders ET. From spatial scale, P, $R_n$, VPD also provide the biggest contributions to ET trend (Figure S3), which positively correlate with their respective trends (Figure S2).

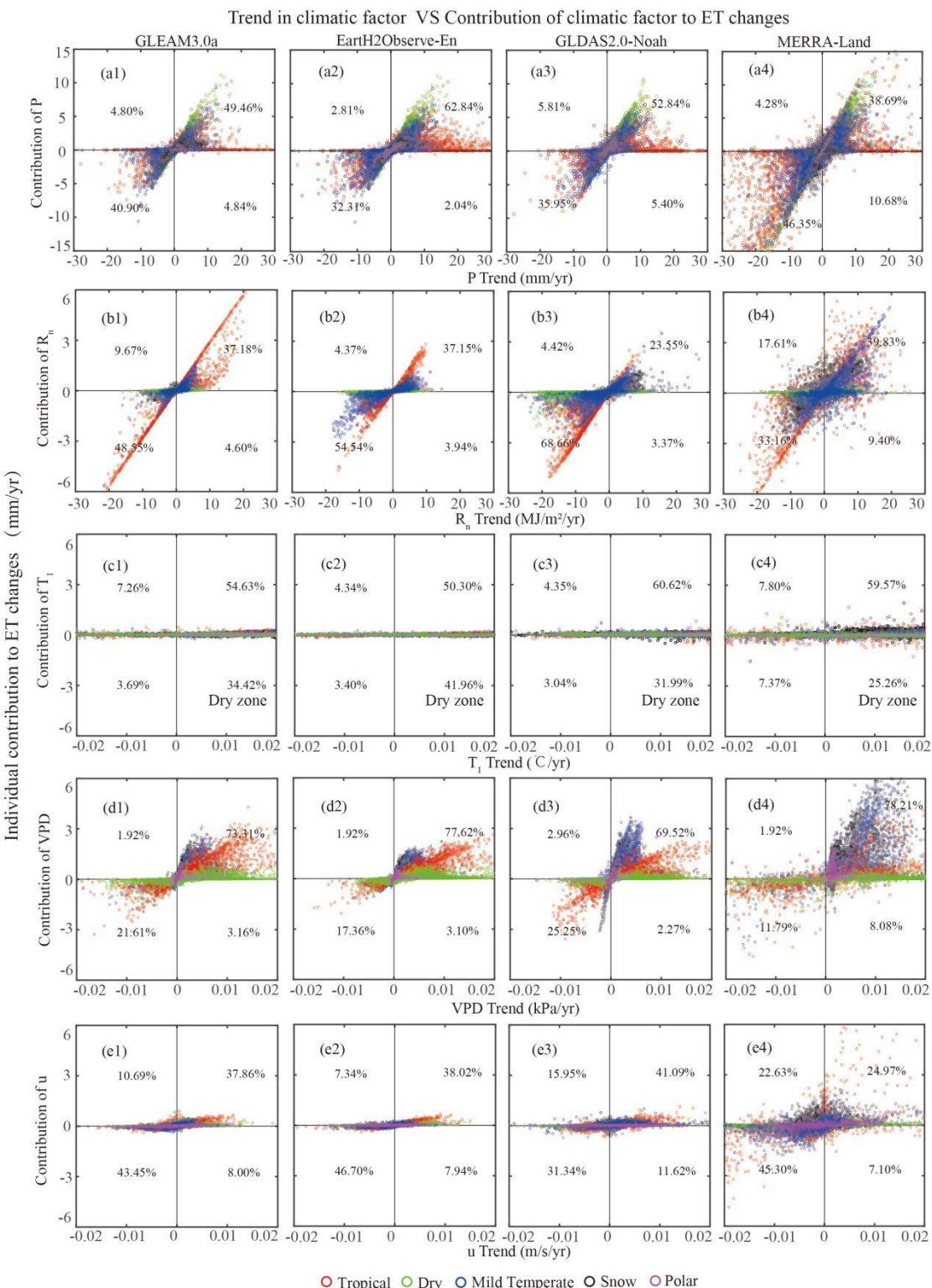

**Figure 3.** Pixel-wise scatterplots of (x-axis) trends in each climatic variable against (y-axis) the contribution of each climatic variable to ET changes. Small letters (a-e) indicate precipitation, radiation, air temperature ($T_1$), VPD and wind speed,

respectively; and numbers (1-4) indicate GLEAM3.0a, EartH2Observe-En, GLDAS2.0-Noah and MERRA-Land respectively. The percentage is the ratio between the number of grid cells in each quadrant and the number of total grid cells; the sum of the percentage values in four quadrants equals to 100%. The color red, green, blue, black purple represents Tropical, Dry, Mild Temperate, Snow, Polar zones, respectively.

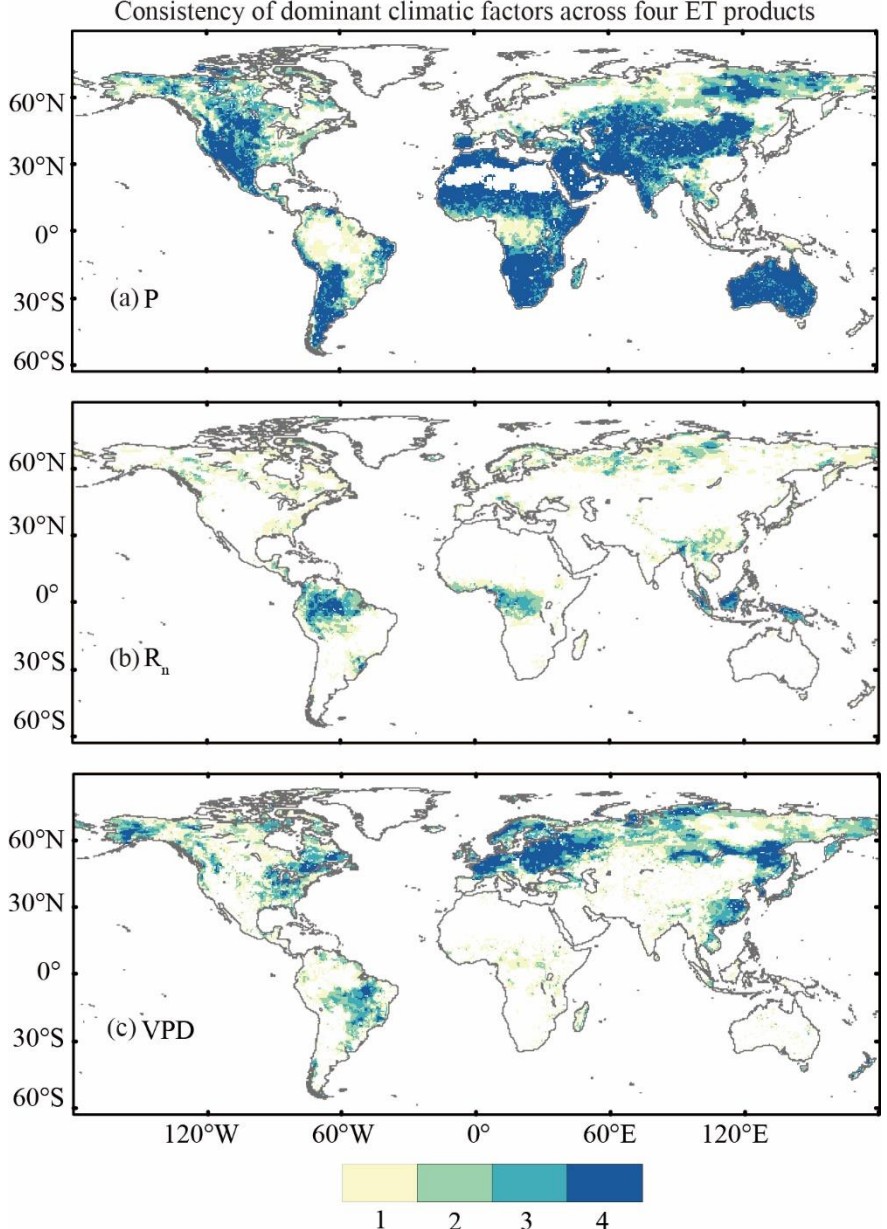

**Figure 4.** The consistency of spatial distribution of dominant climatic factors to global long-term ET trends between GLEAM3.0a, EartH2Observe-En, GLDAS2.0-Noah and MERRA-Land for Precipitation (a), net radiation (b), and VPD (c). The land fraction of air temperature ($T_1$) and wind speed is limited, and the two factors' results are not shown here. Number 1-4 represent the count of these models with the same dominant factor in one pixel, and indicate different confidence level from low to high.

To further show the spatial distribution of these driving factors affecting ET, we compare the consistency of dominant climatic factors across these ET products in Figure 4. The dominant climatic factor is identified with the absolute value of maximum contribution to ET trends. The results indicate that precipitation is the dominant factor of ET trends in the entire Dry zone and some regions of the other climate zones in all models, such as northeastern and southern parts of the Snow zone and the Mild Temperate zone in South America. The net radiation dominates the ET trends in most of the Tropical zone; and VPD dominates the ET trends in the entire Mild Temperate zone, Eastern Europe, and Northeast Asia in the Snow zone. In Table 2, we can see that precipitation, net radiation, and VPD are the dominant factors of ET changes in most global land. For example, precipitation contributes to either positive or negative ET trends in 55.41% of the global grids.

**Table 2.** The percentage of grids in each dominant factor controlling annual ET linear trends for GLEAM3.0a, EartH2Observe-En, GLDAS2.0-Noah, and MERRA-Land. "+" and "-" represent positive, and negative contributions to ET, respectively.

| | | P | $R_n$ | $T_1$ | VPD | u |
|---|---|---|---|---|---|---|
| GLEAM3.0a | + | 30.34% | 4.03% | 0.79% | 27.32% | 0.06% |
| | - | 25.07% | 7.27% | 0.08% | 4.87% | 0.18% |
| EartH2Observe-En | + | 42.70% | 4.65% | 0.44% | 24.88% | 0.06% |
| | - | 19.94% | 4.90% | 0.02% | 2.30% | 0.10% |
| GLDAS2.0-Noah | + | 31.27% | 4.17% | 0.93% | 20.88% | 0.10% |
| | - | 23.67% | 13.73% | 0.01% | 5.17% | 0.07% |
| MERRA-Land | + | 26.77% | 5.60% | 0.08% | 29.09% | 0.06% |
| | - | 30.43% | 6.46% | 0.22% | 0.87% | 0.42% |

## 4 Discussions

### 4.1 Results comparison

In this study, we have found the global ET trends during 1980-2010 in GLEAM3.0a, EartH2Oberve-En, GLDAS2.0-Noah, and MERRA-Land products are relatively consistent. Different ET trends are observed among these products in Africa and South America, where the MERRA-Land shows a significant decrease of about -5.0 mm/yr. A significantly increasing ET pattern is found in some regions of western Europe and southern Asia, the central parts of northern Australia, while a declining ET pattern is observed in western North America and South America. Similar global ET patterns are also found in Pan et al. (2020) based on multi-source products. As shown in Figure 2, there are divergences in the ET trends of the products over some regions. Different ET trends among the products result from different forcing data. For example, MERRA-Land has abnormal negative ET trends over South America and the central part of Africa. This is due to abnormally decreased precipitation providing a negative contribution to ET trends.

How climatic variable controls the global ET trend is one of the crucial questions we ask in this study. We have designed sensitivity experiments to disentangle contributions from each climatic driver (precipitation, net radiation, air temperature, VPD, and wind speed) to answer this question. Precipitation, net radiation, VPD, and wind speed contribute the most to the changes in global ET with an inferred positive relationship. In contrast, an increase in temperature shows the opposite in some regions. The positive relationships between ET and precipitation, net radiation have been confirmed by Lu et al. (2019), Wang et al. (2018b), Pan et al. (2020), and Soni et al. (2021). Precipitation supplies water, and net radiation provides energy for the ET process. However, the increased temperature appears to have influenced the mechanism of ET trend differences between the water-limited region (dry zone) and other regions. Rising air temperature can lead to soil moisture depletion, followed by suppressed vegetation growth in the water-limited regions (Jung et al., 2010; Zhang et al., 2019).

Meanwhile, we found that an increased VPD promotes atmospheric processes followed by global ET changes. Even if increased VPD reduces surface conductance, increased atmospheric water demand by VPD absorbs moisture from soil and vegetation, thus increasing ET (Grossiord et al., 2020). The positive influences are also verified by Kochendorfer et al. (2011); Wang et al. (2012); Yang et al. (2019). However, Novick et al (2016) indicated increased VPD due to global warming increased surface resistance, limiting ET over many biomes. Massmann et al. (2019) suggested that ET response to increased VPD varied from decreasing to increasing, depending on plant water regulation strategies determined by climatic environment and plant types. For example, when compared to boreal and arctic climates, VPD increased ET in tropical and temperate climate; in terms of plant type, shrubs and gymnosperm trees decreased ET, while crops tended to increase ET. Therefore, we consider that the contrasting influence of VPD on ET should be addressed separately, emphasizing how the VPD affects the individual components of ET, which are evaporation from the soil, and canopy interception and transpiration. A more complex physical ET process combined with soil and plant resistance models should be used to do this. For example, Grossiord et al. (2020) admitted that an increased VPD would make stomatal closure, but transpiration would still increase under a certain threshold across plants in different climate regions. Besides, they found the positive response of

surface resistance to VPD increased from wetting to drying climate.

Most studies used the effect of VPD on ET as a surrogate of high air temperature as VPD is determined by air temperature and specific humidity. However, specific humidity changes are weak relative to rapid air warming, resulting in increased VPD controlled by the rising air temperature. Figure 5 shows the spatial pattern of the climatic variables (i.e. air temperature $T_2$ and specific humidity) that dominates the global VPD changes following our proposed sensitivity method. Our study concludes that the specific humidity controls VPD only in some regions of North and South Asia, northern Australia, southern Africa, and South America. However, vegetation physiology controlled by VPD plays a vital role in reshaping the hydrological cycle and global water resources compared to air temperature (Grossiord et al., 2020). Meanwhile, considering a close relationship between T and VPD, we attempted to separate the contribution to ET between VPD and air temperature by designing sim_$T_1$, sim_VPD in Sect. 2.2.2.

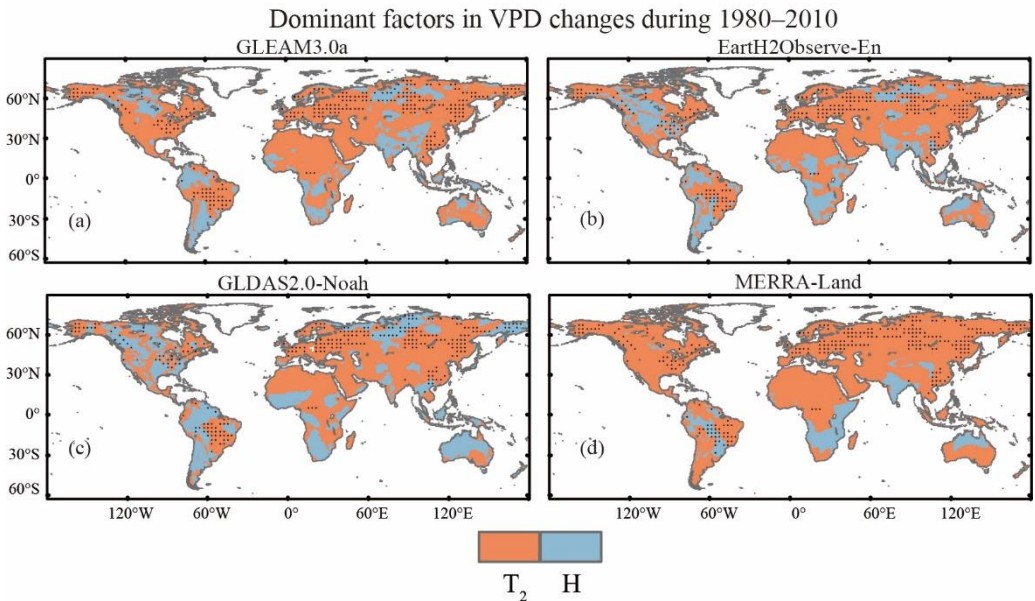

**Figure 5.** Distribution of dominant factor in VPD changes in global land during 1980-2010 for GLEAM3.0a (a), EartH2Observe-En (b), GLDAS2.0-Noah (c), and MERRA-Land (d). $T_2$ and H mean air temperature and specific humidity respectively. Dotted areas mean VPD plays a dominant factor.

Figure 4 shows the spatial distribution of climatic drivers controlling ET implying that precipitation is the primary driver that controls ET in Dry zone. This includes northern and southeastern Eurasia, most of Africa, the middle-west of North America, southern parts of South America, and almost the entire Australia, while net radiation dominates the tropical zone. Why these two climatic drivers are important for controlling ET changes has also attracted interest among other scientists (Pan et al., 2020 and Zhang et al., 2015). Interestingly, we find that the impact of VPD on ET is quite significant in some high latitude regions of the northern hemisphere, such as eastern North America, Europe, and northeastern Asia. Long-term ET changes in these regions are controlled by air temperature (Pan et al., 2020; Zhang et al., 2015). However, in our study, the increased VPD caused by rising air temperature plays a significant role in controlling ET changes (Sottocornola and Kiely, 2010; Kochendorfer et al., 2011; Yang et al., 2019). VPD, rather than air temperature control ET in high latitude regions. Precipitation controls ET in water-deficit regions (Dry zone) by replenishing the storage deficit, and ET in the tropical rain forest (i.e. energy-limited region) is determined by available energy (i.e. net radiation).

## 4.2 Uncertainties

Budyko framework is the key component of the attribution method used in our study. Based on this hypothesis, Fu (1981) and Zhang et al. (2004) offered the best analytical solution (i.e. Sect. 2.2.1 equation (1)) through a dimensional mathematical analysis by providing the mathematical reasons. The key part of the analytical solution is that the ratio between PET and precipitation determines ET. The equation has been applied in numerous hydrological studies at the catchment scale. When

using the hypothesis at the catchment scale, some details of the results related to the land features are often missing or ignored. However, when testing the same hypothesis at grid scales, the Budyko framework performance is outstanding (Greve et al., 2014; Teuling et al., 2019; Roderick et al., 2014). We have also validated the accuracy of the Budyko hypothesis by comparing the ET values estimated by the Budyko with actual ET values in Figure 6. The results with high $R^2$ values for all products indicate the Budyko can be successfully applied in the attribution method. However, there are discrepancies among different PET calculation methods that may introduce some uncertainties into the attribution results. For example, Zhou et al. (2020) compared four temperature-based models (Hamon, Hargreaves-Samani, Oudin, Thornthwaite), two radiation-based models (Energy-Only and Priestley-Taylor), and two synthesis models (Penman and Penman-Monteith) of PET in China as an example, and pointed out that the Penman-Monteith and Penman methods are almost similar but better than the remaining methods. Based on the results, the Penman-Monteith method outperforms the Penman method, thus used to calculate the standard values of PET in our study.

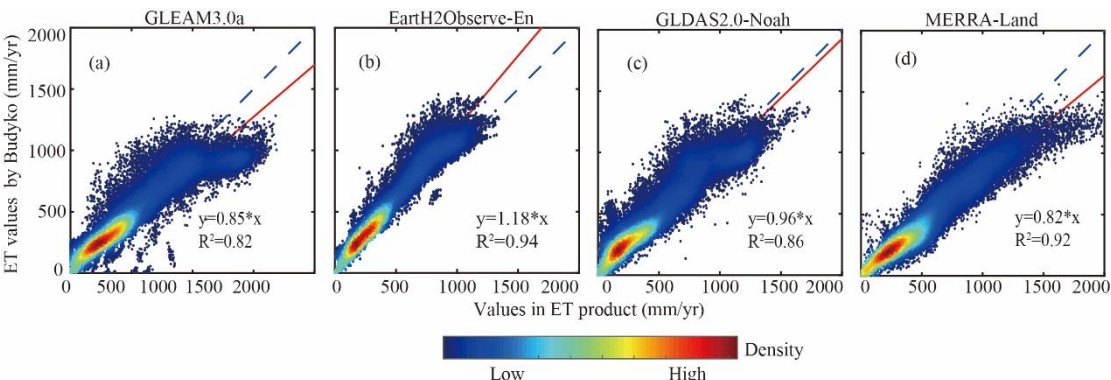

**Figure 6.** Pixel-wise scatterplots of (x-axis) annual ET in each product against (y-axis) annual ET estimated by Budyko Framework. Small letters (a-d) represent GLEAM3.0a, EartH2Observe-En, GLDAS2.0-Noah, and MERRA-Land, respectively.

### 4.2.1 Validations of attribution method

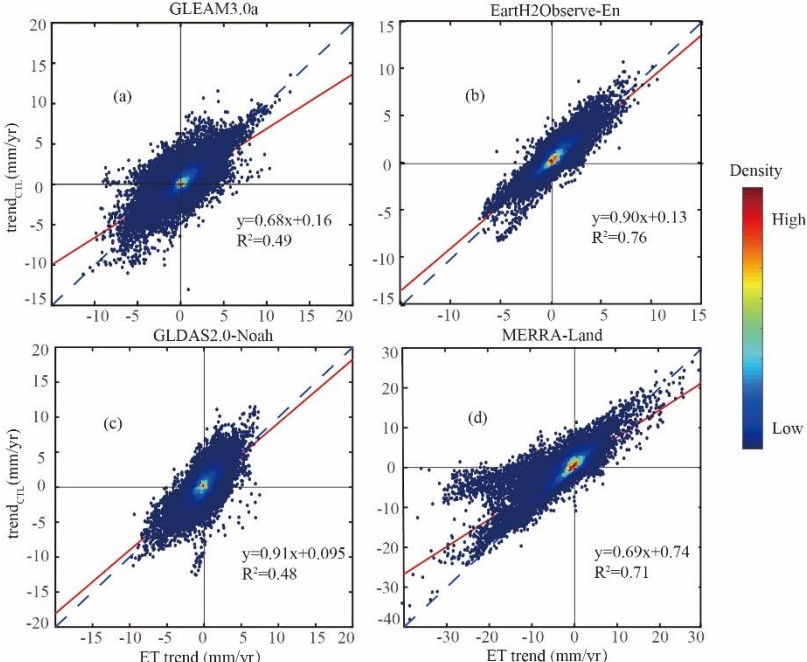

**Figure 7.** The pixel-wise scatterplots of global long-term annual ET linear trend against the control trend (trend$_{CTL}$) in ET for GLEAM3.0a (a), EartH2Observe-En (b), GLDAS2.0-Noah (c), and MERRA-Land (d). The red line indicates a fitted line of the scatter points along with the 1:1 blue dotted line.

The fitted parameter $\underline{\omega}$ in Equ (3) includes landscape characteristics, such as vegetation cover, soil properties, and topography (Xu et al., 2013). The parameter contains each model's characteristic, leading to uncertainties of the attribution

method from forcing data and information in each product. The information consists of structure parameters of each model, and surface factors (land cover types, soil properties, and topography). The selected four products in the study use static surface factors when simulating ET (Table S1). Given the reason, the attribution method is limited to not considering the influences of the land surface on ET changes and only focuses on quantifying several climatic variables' influences here. And, we discuss the influences of vegetation and human activities in next section.

A separation method in this study is used to obtain the respective contribution of each driving factor to the long-term annual ET linear trend, which inevitably is suspected of producing some uncertainties in the attribution results. Figure 7 shows the scatter plots of the pixel-wise ET trend in the four products against those from the CTL experiment to validate the accuracy of reproducing ET with the fitted relations using Equation 3 and the five selected climatic variables. The resultant $R^2$ values range from 0.48 to 0.76, indicating that the $trend_{CTL}$ simulated by Equation 3 can principally reproduce the ET trends as in the four products. Meanwhile, scatter plots of the accumulative contributions of the driving factors ($\sum_{i=1}^{5} C_i$) in each product against the respectively simulated $trend_{CTL}$ are shown in Figure S4, in order to understand the possible uncertainties of such an analysis. Strikingly, the $R^2$ values are all higher than 0.99, indicating that the driving factors' summed contributions are almost equal to the realistic global ET trends in all products.

**4.2.2 Influences of vegetation and human activities**

Vegetation can alter water cycle, and energy cycle by biophysical and biochemical feedback to climate change (Forzieri et al., 2020). For example, global surface greening increases ET/transpiration (Lian et al., 2018; Lu et al., 2021), and reduce soil water content (Li et al., 2018a). However, the complex interaction between vegetation and surface makes it difficult to simulate the influence of dynamic vegetation change on ET (Gentine et al., 2019). Meanwhile, strictly disengaging the contributions of climatic variables and vegetation to ET is very difficult due to the interaction between vegetation and climatic variables (Li et al., 2018b). For water-limited regions, precipitation as main water supply to vegetation controls interannual ET changes (Wang et al., 2021). Thus, the dominating factor of interannual ET changes is not vegetation, but rather, atmospheric climate variables (Zhang et al., 2020). Those studies indicate that contribution of climatic variables have already included information of vegetation, indirectly.

Given the above reasons, the ET products used in this study do not consider the effect of land use /vegetation changes on ET. When simulating ET, the model frameworks assume no interannual land use changes, so they are regarded as static conditions. Detailed landcover types in each product have been shown in Table S1.

Human activities (e.g. irrigation and reservoir construction) have been affecting the components (i.e. ET, runoff, and groundwater storage) of water cycles (Ashraf et al., 2017; Long et al., 2017). For example, the groundwater over North Plain in China, the High Plain in US, and northern India is pumped for agricultural irrigation and contribute to accelerate ET process. Lv et al (2017) indicate that the estimated ET will be more accurate if irrigation water affects hydrological cycles. Unfortunately, most ET products do not consider human activities due to the limited factors of estimated algorithm and model parameters. The GLDAS2.0-Noah and MERRA-Land in this study also do not consider the effect of human activities. GLEAM3.0a partly contains the information of groundwater by considering the effect of ESA-CCI soil moisture on ET. As for EartH2Observe-En, the six models either consider one of groundwater, reservoir, or water use (see Table S1 from Li et al., 2021). However, the attribution results to ET trends in this study show GLEAM3.0a and EartH2Observe-En's validation results are good, indicating that the effect of human activities on ET may be contained in climatic variables. These ET products are produced with appropriate algorithms, parameterizations of models and forcing data sets. The accuracy of ET has been validated by the respective developers; Li et al (2018) in China, Wang et al (2018) in Yellow River basin, and Nooni et al (2019) in Nile River basin, suggesting good performances of these products. Therefore, our study only focuses on climatic factors affecting interannual ET changes. For future studies, the contribution of land surface such as human

activities to ET should be investigated to understand the mechanism of global ET trend better. Additionally, we only consider local contributions of ET here. In fact, large-scale modes of climate variability (e.g. El Niño Southern Oscillation, the North Atlantic Oscillation,) can also affect terrestrial evaporation. For example, Martens et al (2018) indicate that El Niño Southern Oscillation controls the overall dynamic of global land ET, while some models dominate regional ET change, such as East Pacific–North Pacific teleconnection patterns.

### 4.2.3 The relationship between fitted parameter ω and ET trend analysis, vegetation

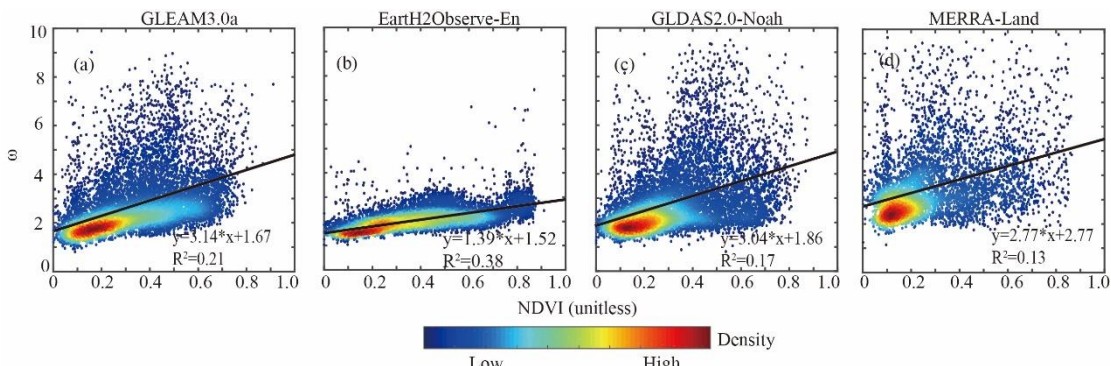

**Figure** 8. Pixel-wise scatterplots of (x-axis) multi-year average NDVI against (y-axis) their fitted ω values in each product. Small letters (a-d) represent GLEAM3.0a, EartH2Observe-En, GLDAS2.0-Noah, and MERRA-Land. GIMMS NDVI data during 1982-2010 is used here.

Here, we compare ET trends in each product to climate zones, which are represented by aridity index. The aridity index (PET/precipitation) in each product is calculated with respective precipitation and PET data. Figure S5(a1-d1) show that the biggest ET trends of all products exit the wettest regions (low aridity index). To study the influence of fitted parameter ω on ET trend analysis, we compare the control on ET trend (trendCTL) to the aridity index. The results in Figure S5(a2-d2) show similar results to the actual ET trend, meaning the ET trend analysis in the attributed method can capture actual ET change characteristics. Meanwhile, we also quantify the relationship of parameter ω fitted by precipitation, potential evapotranspiration, and actual evapotranspiration in each product to multi-year average GIMMS NDVI during 1982-2010. Figure 8 shows the linear relationship between fitted parameter ω and NDVI for all products with $R^2$ values of 0.13-0.38. In general, parameter ω can be calculated according to the linear relationship between ω and NDVI (Bai et al, 2019; Greve et al, 2014). The results show that our trend analysis keeps the relationship, spatially. However, we admit that time-varying ω (e.g. vegetation, soil property) will directly affect ET (Lu et al., 2021). The impact of ω would vary as a function of the chosen timescale which requires a more indepth study beyond the scope of the current study.

### 5. Conclusions

We have estimated the linear ET trend globally during 1980-2010 from GLEAM3.0a, EartH2Oberve-En, GLDAS2.0-Noah, and MERRA-Land. Secondly, we obtained the respective contribution of each factor to ET trends with multiple sensitivity experiments and a separation method and identified which factor controls global ET changes across different climate zones. The major findings are summarized below:

(1) ET changes: Long-term trend in ET during 1980-2010 is evident globally, especially in Africa and South America. A significant increase in ET is observed in Eurasia, northern and central Australia, northeast Africa, eastern parts of South America, and eastern parts of central North America. Decreasing ET is found in the west of North and South America, northeast of Africa, and the Arabian Peninsula. MERRA-Land has more significant ET changes when compared to the other products.

(2) Dominant factors: Precipitation, net radiation, VPD, and wind speed are positively correlated to global ET changes, while air temperature ($T_1$) has contrast influences on ET between dry zone and other regions.

Precipitation controls ET changes in Dry zone, including north, and central and southeast regions of Eurasia, most of Africa, central parts of western North America, southern parts of South America, and almost entire Australia, net radiation dominates the tropical zone. VPD dominates ET in some high latitude regions of the northern hemisphere, such as eastern North America, the whole of Europe, and northeastern Asia.

(3) Uncertainties of ET trends: Global ET trends among the products are determined by their climate variables. Different sources of forcing data sets result in different magnitudes of ET trends, even the reversing signs. But consistent above attribution results in those products confirm that ET mechanisms are robust.

**Data availability.** In this study, each ET global product and respective forcing climatic factors can be downloaded: GLEAM 3.0a from https://www.gleam.eu/, EartH2Observe-En from http://www.earth2observe.eu/., GLDAS2.0-Noah from https://disc.gsfc.nasa.gov/datasets?keywords=GLDAS,and MERRA-Land from https://disc.gsfc.nasa.gov/datasets?keywords=merra-land&page=1.

**Author Contributions.** Shijie Li, Guojie Wang, Jian Peng: Conceptualization, Methodology. Shijie Li, Guojie Wang, Chenxia Zhu: Data curation, Writing- Original draft preparation. Jiao Lu, Waheed Ullah, Daniel Fiifi Tawia Hagan, Giri Kattel: Writing-Reviewing and Editing. Guojie Wang, Jian Peng: Supervision.

**Competing Interest.** The authors declare that they have no known competing financial interests or personal relationships that could have appeared to influence the work reported in this paper.

**Acknowledgments**. This research was funded by National Key Research and Development Program of China (#2017YFA06 03701), the National Natural Science Foundation of China (#41875094), the Sino-German Cooperation Group Project (#GZ1 447), and Postgraduate Research and Practice Innovation Program of Jiangsu Province (#KYCX20_0932).

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
