# Peer review of "Attribution of global evapotranspiration trends based on the Budyko framework"

_Hydrology and Earth System Sciences, 2021_

## Author Comment (AC1)

Reviewer 1:

Accurate quantification of the climatic contributions for global land evapotranspiration change is necessary for understanding variability in the global water cycle. This study assembled four ET datasets based on various methodological sources, further adopted the Budyko framework and sensitivity experiment analysis to quantifying the contribution of climatic variables (P, Rn, T, VPD and u) to ET trend. The analysis identified the main climatic factor controls ET trend on a global scale. This research is systemic and detailed, helps reveal the controlling factors of global ET change. The main comments can be found as follows:

**Response:** We thank this reviewer for constructive comments, which significantly improves the quality of the manuscript. The following is our detailed responses to the reviewer's comments.

1.The expression should be improved.

**Response:** We have improved unclear expressions in the manuscript.

2.Budyko method was used to conduct a control experiment to compare with ET product results, and the ω parameters were obtained by least squares fitting, did the authors use annual data for the entire period for the fitting? If this were the case, it would not be possible to consider the effect of land use changes on the ω parameters and thus bias the estimated ET simulations, especially considering that such a long study period (1980-2010) with significant land use changes must have an important impact on ω.

**Response:** We appreciate this comment.

In this study, ω parameters are fitted with annual precipitation, potential evapotranspiration, and actual evapotranspiration. The parameters ω in Budyko framework are determined by landscape characteristics (e.g. vegetation cover, soil properties, and topography) (Yang et al., 2008), in particular ω, which parameters can be related to vegetation changes (Greve et al., 2014). As the reviewer pointed out, land-use changes during such a long study period (1980-2010) significantly affect evapotranspiration as a function of ω. For example, vegetation greening is indicated to control interannual evapotranspiration variation (Lu et al., 2021). However, all the four ET data used in this paper assume no interannual vegetation changes (satellite phenology driven), when simulating ET(detailed landcover types in each product have been shown in Table S1). It is worth noting that the assumption mentioned above does not discredit the reliability of the ET products. Furthermore, the contributions of climatic variables to ET trends already include information of the vegetation

indirectly. The accuracy of the ET products has been validated in different studies such as Li et al (2018) in China, Wang et al (2018) over the Yellow River basin, and Nooni et al (2019) in the Nile River basin, among others, suggesting good performances of these products. Therefore, our study only focuses on climatic factors affecting interannual ET changes.

We have added discussions about vegetation changes in the manuscript. The text there reads as: "*Vegetation can alter water cycle, and energy cycle by biophysical and biochemical feedback to climate change (Forzieri et al., 2020). For example, global surface greening increases ET/transpiration (Lian et al., 2018; Lu et al., 2021), and reduce soil water content (Li et al., 2018a). However, the complex interaction between vegetation and surface makes it difficult to simulate the influence of dynamic vegetation change on ET (Gentine et al., 2019). Meanwhile, strictly disengaging the contributions of climatic variables and vegetation to ET is very difficult due to the interaction between vegetation and climatic variables (Li et al., 2018b). For water-limited regions, precipitation as main water supply to vegetation controls interannual ET changes (Wang et al., 2021). Thus, the dominating factor of interannual ET changes is not vegetation, but rather, atmospheric climate variables (Zhang et al., 2020). Those studies indicate that contribution of climatic variables have already included information of vegetation, indirectly.*

*Given the above reasons, the ET products used in this study do not consider the effect of land use /vegetation changes on ET. When simulating ET, the model frameworks assume no interannual land use changes, so they are regarded as static conditions. Detailed landcover types in each product have been shown in Table S1*".

Table S1. Comparisons of landcover types data used by the four ET products

| ET product | | Landcover types data | Period |
|---|---|---|---|
| GLEAM3.0a | | MOD44B | Static |
| GLDAS2.0-Noah | | MCD12Q1 | Static |
| MERRA-Land | | Global Land Cover Characterization | Static |
| | W3RA | MOD44B | Static |
| | HTESSEL-CaMa | | Static |
| | JULES | Global Land Cover Characterization | Static |
| EartH2Observe-En | PCR-GLOBWB | | Static |
| | LISFLOOD | GlobCover2009 | Static |
| | HBV-SIMREG | | Static |
| | WaterGAP3 | MOD12Q1 | Static |

Note: However, regarding EartH2Observe-En, the LUC datasets used by seven (in this table) and two models (i.e., ORCHIDEE and SURFEX-TRIP) are available and unavailable, respectively; the LUC is not the necessary input for SWBM.

**References:**

Forzieri, Giovanni; Miralles, Diego G.; Ciais, Philippe; Alkama, Ramdane; Ryu, Youngryel; Duveiller, Gregory; Zhang, Ke; Robertson, Eddy; Kautz, Markus; Martens, Brecht; Jiang, Chongya; Arneth, Almut; Georgievski, Goran; Li, Wei; Ceccherini, Guido; Anthoni, Peter; Lawrence, Peter; Wiltshire, Andy; Pongratz, Julia; Piao, Shilong; Sitch, Stephen; Goll, Daniel S.; Arora, Vivek K.; Lienert, Sebastian; Lombardozzi, Danica; Kato, Etsushi; Nabel, Julia E. M. S.; Tian, Hanqin; Friedlingstein, Pierre; Cescatti, Alessandro. Increased control of vegetation on global terrestrial energy fluxes. Nat. Clim. Chang. 2020, 10, 356–362.

Gentine P, Green J K, Guerin M, et al. Coupling between the terrestrial carbon and water cycles - a review. Environmental Research Letters, 2019, 14(8).

Greve P, Orlowsky B, Mueller B, et al. Global assessment of trends in wetting and drying over land. Nature Geoscience, 2014, 7(10):716-721.

Li, S.J., Wang, G.J., Sun, S.L., Chen, H.S., et al., 2018. Assessment of Multi-Source Evapotranspiration Products over China Using Eddy Covariance Observations. Remote Sensing, 210(11), 1692.

Lian, X., Piao, S., Huntingford, C., Li, Y., Zeng, Z., Wang, X., Wang, T., 2018. Partitioning global land evapotranspiration using CMIP5 models constrained by observations. Nature Climate Change 8 (7), 640–646.

Lu, J., Wang, G., Li, S., Feng, A., Zhan, M., Jiang, T., et al. (2021). Projected land evaporation and its response to vegetation greening over China under multiple scenarios in the CMIP6 models. Journal of Geophysical Research: Biogeosciences, 126, e2021JG006327.

Lu, J., Wang, G., Li, S., Feng, A., Zhan, M., Jiang, T., et al. (2021). Projected land evaporation and its response to vegetation greening over China under multiple scenarios in the CMIP6 models. Journal of Geophysical Research: Biogeosciences, 126, e2021JG006327.

Nooni, I.K.; Wang, G.; Hagan, D.F.T.; Lu, J.; Ullah, W.; Li, S. Evapotranspiration and its Components in the Nile River Basin Based on Long-Term Satellite Assimilation Product. Water 2019, 11, 1400.

Wang, G.J., Pan, J., Shen, C.C., Li, S.J., Lu, J., Lou, D., Hagan, D. F. T., et al., 2018. Evaluation of Evapotranspiration Estimates in the Yellow River Basin against the Water Balance Method. Water, 10(12):1884.

Wang, H.N., Lv, X.Z., Zhang, M.Y., 2021. Sensitivity and attribution analysis of vegetation changes on evapotranspiration with the Budyko framework in the Baiyangdian catchment, China. Ecol. Indic. 120, 106963.

Y. Li, S. Piao, L. Z. X. Li, A. Chen, X. Wang, P. Ciais, L. Huang, X. Lian, S. Peng, Z. Zeng, K. Wang, L. Zhou, Divergent hydrological response to large-scale afforestation and vegetation greening in China. Sci. Adv., 2018a, 4, eaar4182.

Yang, H., Yang, D., Lei, Z., & Sun, F. (2008). New analytical derivation of the mean annual water-energy balance equation. Water Resources Research, 44(3), W03410.

Yue Li, Zhenzhong Zeng, Ling Huang, Xu Lian and Shilong Piao. Science. Comment on "Satellites reveal contrasting responses of regional climate to the widespread greening of Earth". Science, 2018b, 360 (6394), eaap7950.

Zhang, D., Liu, X., Zhang, L., Zhang, Q., Gan, R., Li, X., 2020. Attribution of evapotranspiration changes in humid regions of China from 1982 to 2016. J. Geophys. Res.: Atmos., 125 (13), e2020JD032404.

3. Figure 7: As the percentage of grids in each dominant factor controlling annual ET linear trends has been distinguished in Table 2, I suggest to focus on the regions where VPD plays a dominant factor in Figure 7.

**Response:** Thank you for your comment.

We have highlighted the regions where VPD plays a dominant factor with the dotted areas in Figure 7.

[Figure]

**Figure 7.** Distribution of dominant factor in VPD changes in global land during 1980-2010 for GLEAM3.0a (a), EartH2Observe-En (b), GLDAS2.0-Noah (c), and MERRA-Land (d). $T_2$ and H mean air temperature and specific humidity respectively. Dotted areas are where VPD is a dominant factor.

4. 3.3 Validations of attribution method belongs to the 4.2 Uncertainties, as this section discusses the reliability of Budyko method in ET estimation and attribution analysis.

**Response:** Thank you for your comment.

We have added section 3.3 Validations of attribution method into section 4.2.1 Validations of attribution method.

5. Abstract Line 22: "land-atmosphere interactions" & Page 10 Line 24: "The positive feedbacks": The main conclusion of this article is demonstrating the main factor affecting ET trend. However, it appears that this study did not address the interaction or feedback between ET and VPD.

**Response:** Thank you for your comment.

Abstract Line 22: "land-atmosphere interactions" has been changed to "carbon-water-energy cycle".

Page 10 Line 24: "The positive feedbacks" has been changed to "The positive influences".

6. As the authors mentioned choice of ET data may add significant uncertainties into the ET attribution. The authors need to show how the impact of the results due to ET datasets uncertainty is reduced and summarize the combined results from multiple data sets, rather than one data set with one result without giving a combined conclusion. And this should also be summarized in Conclusion.

**Response:** Thank you for your comment.

It is challenging to study ET change mechanisms only depending on one product. Because the model structures, algorithms, and forcing data sets can affect ET accuracy (Martens et al., 2017), when simulating ET. Therefore, we decided to use multi-source ET products and their forcing data sets. As described in Figure 2, there are evident differences in ET trends among those products. Different trends of climatic variables can directly affect ET trends. Figure S2 shows the spatial distribution of the annual linear trend in each driving factor (i.e., P, $R_n$, T, VPD, and u) during 1980-2010. We can find that precipitation and net radiation have differences between the products, especially for precipitation trends in MERRA-Land and net radiation trends in GLDAS2.0-Noah. By the attribution method with Budyko framework, the global long-term annual ET linear trend responses to climatic variables' changes can be quantified in Figure S3. Compared to air temperature and wind speed, precipitation, net radiation, VPD provide the biggest contribution to ET trends. As the reviewer said, we need to summarize ET conclusions from different products' results. To do this, we obtain the consistency of the dominant factor in ET trends among those products by summarizing the results in Figure 4.

Similar descriptions in this manuscript can be added. The text there reads as: "Spatially, *P, $R_n$, VPD also provide the biggest contributions to ET trend (Figure S3), which are positively correlated with their respective trends (Figure S2)*". Meanwhile, we summarize this in Conclusion, and the text there reads as "Global ET trends among the products are determined by their climate variables. Different sources of forcing data sets result in different magnitudes of ET trends, even the reversing signs. But consistent above attribution results in those products confirm that ET mechanisms are robust".

[Figure]

Figure S2. Spatial distribution of annual linear trend in each driving factor during 1980-2010. Small letters (a-e) respectively indicate P, Rn, T, VPD, and u and numbers (1-4) represent GLEAM3.0a, EartH2Observe-En, GLDAS2.0-Noah, and MERRA-Land. Dotted area indicates the trend passes significance level (p<0.05).

[Figure]

Figure S3. Attributions of the global long-term annual ET linear trend during 1980-2010. Small letters (a-e) indicate P, Rn, T, VPD and u respectively; and numbers (1-4) indicate the ET products of GLEAM3.0a, EartH2Observe-En, GLDAS2.0-Noah and MERRA-Land respectively.

**References:**

Martens, B., Miralles, D.G., Lievens, H., van der Schalie, R., de Jeu, R.A.M., Fernández-Prieto, D., Beck, H.E.,Dorigo,W.A., Verhoest, N.E.C.: GLEAM v3: Satellite-based land evaporation and root-zone soil moisture. Geosci. Model Dev. 10, 1903–1925, 10.5194/gmd-10-1903-2017, 2017.

7. Table 2 gives the percentage of grids in each dominant factor controlling annual ET linear trends with positive and negative. Meanwhile, Figure 2 shows the spatial distribution of annual ET linear trends for 4 datasets, opposite trends between different products in the same pixel can be found. My concern is whether the areas with positive ET trend in one dataset are changing negatively in the other dataset.

**Response:** Thank you for your comment.

As shown in Figure 2, there are divergences in the ET trends of the products over some regions. Different ET trends among the products result from different forcing data (Table 1). Each climatic factor's contribution to ET trends in Figure S3 is determined by the respective factor's trend in Figure S2. For example, MERRA-Land has abnormal negative ET trends over South America and the central part of Africa. By comparing Figure S1 with Figure S2, we find that negative ET trends over the central part of Africa are due to abnormally decreased precipitation providing a negative contribution to ET trends. Similar description has been added, and The text there reads as: "*As shown in Figure 2, there are divergences in the ET trends of the products over some regions. For example, MERRA-Land has abnormal negative ET trends over South America and the central part of Africa. This is due to abnormally decreased precipitation providing a negative contribution to ET trends*".

Some specific comments:

1 Page 1, Line 25: As you mentioned "terrestrial water flux component", "accounting for more than 60% of global precipitation" should be "land precipitation".

**Response:** Thank you for your comment.

We have changed "global precipitation" to "global land precipitation".

2 Page 3, 2.1 Data: Forcing data in Budyko framework and Köppen climate classification should also be summarized.

**Response:** Thank you for your comment.

We have added the description, liking "*In the attribution method with Budyko framework, we use respective forcing data of each product (please see detail*

*description in section 2.2 Forcing data)*"; and "*The Köppen climate classification is produced according to the empirical relationship between climatic variables and vegetation*".

3 Page 5, Line 35: What's the meaning of Ci?

**Response:** Thank you for your comment.

Ci means the contribution of each factor to ET change in each product.

4 Figure 4: The image color scheme can be more distinguishable.

**Response:** Thank you for your comment.

Figure 4's color scheme has been changed:

[Figure]

Figure 4. The consistency of spatial distribution of dominant climatic factors to global long-term ET trends between GLEAM3.0a, EartH2Observe-En, GLDAS2.0-Noah and MERRA-Land for Precipitation (a), net radiation (b), and VPD (c). The land fraction of air temperature (T1) and

wind speed is limited so their results are not shown here. Numbers 1-4 represent the count of these models with the same dominant factor in one pixel, and indicate different confidence level from low to high.

5 Page 5, Line 10: How do you define the "dominant factor of ET trends"? Please give an explanation or algorithm.

**Response:** Thank you for your comment.

We have added the explanation. The text there reads as: "The dominant climatic factor is identified with the absolute value of maximum contribution to ET trends among those factors".

6 Figure 5 & 8: Please use density scatter plot to improve image quality.

**Response:** Thank you for your comment.

Figure 5 & 8's color scheme have been changed:

[Figure]

Figure 5. The pixel-wise scatterplots of global long-term annual ET linear trend against the control trend (trendCTL) in ET for GLEAM3.0a (a), EartH2Observe-En (b), GLDAS2.0-Noah (c), and MERRA-Land (d). The red line indicates a fitted line of the scatter points along with the 1:1 blue dotted line.

[Figure]

Figure 8. Pixel-wise scatterplots of (x-axis) annual ET in each product against (y-axis) annual ET estimated by Budyko Framework. Small letters (a-d) represent GLEAM3.0a, EartH2Observe-En, GLDAS2.0-Noah, and MERRA-Land, respectively.

7 Please avoid citing a large number of references in one place.

**Response:** Thank you for your comment.

We have deleted some unnecessary references in one place.

---

## Author Comment (AC2)

Reviewer 2:

The manuscript "Attributing of global evapotranspiration trends based on the Budyko framework" by Li et al. investigated the trend of evapotranspiration (ET) at global scale and its contributing factors, including precipitation (P), net radiation (Rn), air temperature (T1), VPD, and wind speed (u), by using multiple datasets (GLEAM3.0a, EartH2Observe ensemble, GLDAS2.0-Noah and MERRA-Land). The methods and datasets used in this study is similar to a previous study (Li. et al., Journal of Hydrology, 2021) by the same author except this manuscript extends previous study in China to global. This study is more like a numerical sensitivity exercise, suffers from methodological methodological flaws and does not provide insights to understand ET trend and its contributing factors.

**Response:** We thank this reviewer for constructive comments about the accuracy of the attribution method in our work, which significantly improves the quality of the manuscript. The following is our detailed responses to the reviewer's comments. Compared to JH et al., 2021, we discuss evident differences in results and uncertainties of the attribution method in this manuscript.

For result differences, "*Li et al (2021) attempted to quantify the contribution of those forcing variables to ET trends over China with the Budyko theory. Their study indicates that precipitation dominates ET trends over water-limited regions, while VPD controls ET of energy-limited regions. However, there are still unclear questions about the global land ET mechanism. For example, how differently would the conclusions of dominating ET factors over water-limited regions be for global dry lands? Which variable controls ET over the global tropical zone is unclear, despite the results of VPD controlling ET over the energy-limited region of China. Wang et al (2022) indicate that global significantly increased ET mostly results from increasing air temperature, especially in the humid region. Pan et al (2020) point out that precipitation, air temperature, and radiation control Amazon's ET changes. On the other hand, the boreal ET mechanisms are also not entirely clear. Increasing air temperature is significantly correlated with ET (Wang et al., 2022), while increasing VPD contributes to ET process over the boreal region (Helbig et al., 2020). Therefore, it is necessary to assess global ET mechanisms using the same attribution method for solving these problems*".

For uncertainties of the attribution method, we quantify the contribution of air temperature T2 and specific humidity to global VPD changes following our proposed sensitivity method in Figure 5. Our study concludes that the specific humidity

controls VPD only in some regions of North and South Asia, northern Australia, southern Africa, and South America. We also discuss the relationship between fitted parameter ω and ET trend analysis, vegetation. The text there reads as: "Here, we compare ET trends in each product to climate zones, *which are represented by aridity index. The aridity index (PET/precipitation) in each product is calculated with respective precipitation and PET data. Figure S5(a1-d1) show that the biggest ET trends of all products exit the wettest regions (low aridity index). To study the influence of fitted parameter ω on ET trend analysis, we compare the control on ET trend (trend$_{CTL}$) to the aridity index. The results in Figure S5(a2-d2) show similar results to the actual ET trend, meaning the ET trend analysis in the attributed method can capture actual ET change characteristics. Meanwhile, we also quantify the relationship of parameter ω fitted by precipitation, potential evapotranspiration, and actual evapotranspiration in each product to multi-year average GIMMS NDVI during 1982-2010. Figure 8 shows the linear relationship between fitted parameter ω and NDVI for all products with R2 values of 0.13-0.38. In general, parameter ω can be calculated according to the linear relationship between ω and NDVI (Bai et al, 2019; Greve et al, 2014). The results show that our trend analysis keeps the relationship, spatially. However, we admit that time-varying ω (e.g. vegetation, soil property) will directly affect ET (Lu et al., 2021). The impact of ω would vary as a function of the chosen timescale which requires a more indepth study beyond the scope of the current study*".

**References:**

Helbig, M., Waddington, J.M., Alekseychik, P. et al. Increasing contribution of peatlands to boreal evapotranspiration in a warming climate. Nat. Clim. Chang. 10, 555–560 (2020).

Pan, S., Pan, N., Tian, H., Friedlingstein, P., Sitch, S., Shi, H., Arora, V. K., Haverd, V., Jain, A. K., Kato, E., Lienert, S., Lombardozzi, D., Nabel, J. E. M. S., Ottlé, C., Poulter, B., Zaehle, S. N. & Running, S. W.: Evaluation of global terrestrial evapotranspiration using state-of-the-art approaches in remote sensing, machine learning and land surface modeling. Hydrology and Earth System Sciences, 24, 1485-1509, 10.5194/hess-24-1485-2020, 2020.

Wang Ren et al. Recent increase in the observation-derived land evapotranspiration due to global warming. Environ. Res. Lett. 17 (2022) 024020.

Major comments

1. The Budyko equation assumes that precipitation is the only water supply for ET. At global scale during the study period (1980-2010), many regions have experienced long-term trends in groundwater storage. For example, in many regions (e.g., the

North Plain in China, the High Plain in US, the northern India) where groundwater is used for agricultural irrigation, the depleted groundwater provides an additional source for ET. In this study, both the analytical framework (Budyko equation) and some of the datasets (e.g., GLDAS2-Noah) do not capture groundwater dynamics. Therefore, this study only investigated the climatic factors on ET trend and cannot provide a full picture of ET trend. Even if the ET trend caused by groundwater is captured (e.g., by the remote sensing based GLEAM ET product), this manuscript may mistakenly attribute ET trend caused by groundwater to climatic factors.

**Response:** Thank you for your comments.

We agree that human activities (irrigation and reservoir construction) have been playing an important role in water cycle. To discuss that, we add a new section. The text there reads as: such as "*Human activities (e.g. irrigation and reservoir construction) have been affecting the components (i.e. ET, runoff, and groundwater storage) of water cycles (Ashraf et al., 2017; Long et al., 2017). For example, the groundwater over North Plain in China, the High Plain in US, and northern India is pumped for agricultural irrigation and contribute to accelerate ET process. Lv et al (2017) indicate that the estimated ET will be more accurate if irrigation water affects hydrological cycles. Unfortunately, most ET products do not consider human activities due to the limited factors of estimated algorithm and model parameters. The GLDAS2.0-Noah and MERRA-Land in this study also do not consider the effect of human activities. GLEAM3.0a partly contains the information of groundwater by considering the effect of ESA-CCI soil moisture on ET. As for EartH2Observe-En, the six models either consider one of groundwater, reservoir, or water use (see Table S1 from Li et al., 2021). However, the attribution results to ET trends in this study show GLEAM3.0a and EartH2Observe-En's validation results are good, indicating that the effect of human activities on ET may be contained in climatic variables. These ET products are produced with appropriate algorithms, parameterizations of models and forcing data sets. The accuracy of ET has been validated by the respective developers; Li et al (2018) in China, Wang et al (2018) in Yellow River basin, and Nooni et al (2019) in Nile River basin, suggesting good performances of these products. Therefore, our study only focuses on climatic factors affecting interannual ET changes. For future studies, the contribution of land surface such as human activities to ET should be investigated to understand the mechanism of global ET trend better. Additionally, we only consider local contributions of ET here. In fact, large-scale*

*modes of climate variability (e.g. El Niño Southern Oscillation, the North Atlantic Oscillation,) can also affect terrestrial evaporation. For example, Martens et al (2018) indicate that El Niño Southern Oscillation controls the overall dynamic of global land ET, while some models dominate regional ET change, such as East Pacific–North Pacific teleconnection patterns*".

Therefore, the direct contributions of ground water and soil moisture are not considered, although we are aware that they do play a role since we mainly focus on atmospheric factors. Additionally, water transport from the ocean and other sources (remote sources) such as shown in Wei et al. (2012, 2016) are also not considered. The goal was to simplify this whole framework and then in following studies, we also look into the impact of land and other remote sources.

Table S1. Members of Eearth2Observe-En ET product of considering human activities (ground water, reservoir lakes, and water use) (Li et al., 2021).

| Name | Ground water | Reservoir/ Lakes | Water use |
|---|---|---|---|
| HTESSEL-CaMa | NO | NO | NO |
| JULES | NO | NO | NO |
| LISFLOOD | YES | YES | YES |
| ORCHIDEE | YES | NO | Irrigation only |
| PCR-GLOBWB | YES | Only lakes | NO |
| SURFEX-TRIP | YES | NO | NO |
| HBV-SIMREG | NO | NO | NO |
| SWBM | NO | NO | NO |
| W3RA | YES | NO | NO |
| WaterGAP3 | YES | YES | YES |

Note: HTESSEL-CaMa is Hydrology Tiled ECMWF Scheme for Surface Exchanges over Land-Catchment-based Macro-scale Floodplain model; JULES is the Joint UK Land Environment Simulator; PCR-GLOBWB is PCRaster GLOBal Water Balance model; HBV-SIMREG is Hydrologiska Byråns Vattenbalansavdelning model; SWBM is Simple Water Balance Model; W3RA is Water Resources Assessment; WaterGAP3 is Water-Global Assessment and Prognosis. Detail information refers to Schellekens et al. (2017).

**References:**
Martens, Brecht; Waegeman, Willem; Dorigo, Wouter A.; Verhoest, Niko E. C.; Miralles, Diego G. (2018). Terrestrial evaporation response to modes of climate variability. npj Climate and Atmospheric Science, 1(1), 43.
Wei, J. and P. A. Dirmeyer, 2012: Dissecting soil moisture-precipitation coupling, Geophysical Research Letters, 39, L19711, https://doi.org/10.1029/2012GL053038.
Wei, J., H. Su, and Z.-L. Yang, 2016. Impact of moisture flux convergence and soil moisture on precipitation: a case study for the southern United States with implications for the globe, Climate Dynamics, 46, 467-481.
Ashraf, B., AghaKouchak, A., Alizadeh, A., et al., 2017. Quantifying anthropogenic stress on groundwater resources. Sci. Rep. 7 (1), 12910.

Li, S., Wang, G., Sun, S., et al.: Long-term changes in evapotranspiration over China and attribution to climatic drivers during 1980-2010. Journal of Hydrology. 595(1-4):126037. 10.1016/j.jhydrol.2021.126037, 2021.

Long, D., Pan, Y., Zhou, J., Chen, Y., Hou, X.Y., Hong, Y., Scanlon, B.R., & Longuevergne, L. (2017). Global analysis of spatiotemporal variability in merged total water storage changes using multiple GRACE products and global hydrological models. Remote Sensing of Environment, 192, 198-216.

Lv, M., Ma, Z., Yuan, X., et al., 2017. Water budget closure based on GRACE measurements and reconstructed evapotranspiration using GLDAS and wateruse data for two large densely-populated mid-latitude basins. J. Hydrol. 547, 585–599.

Martens, B., Miralles, D.G., Lievens, H., van der Schalie, R., de Jeu, R.A.M., Fernández-Prieto, D., Beck, H.E.,Dorigo,W.A., Verhoest, N.E.C.: GLEAM v3: Satellite-based land evaporation and root-zone soil moisture. Geosci. Model Dev. 10, 1903–1925, 10.5194/gmd-10-1903-2017, 2017.

Schellekens, J., Dutra, E., Martínez-de la Torre, A., Balsamo, G., van Dijk, A., et al., 2017. A global water resources ensemble of hydrological models: The eartH2Observe Tier-1 dataset. Earth Syst. Sci. Data. 9, 389–413.

2. The parameter w in Budyko equation in Equation 1 is obtained by regression using each set of data product (Line 7-8). I assume that the authors repeat the regression four times using the four sets of P, PET and ET data. The parameter w is usually associated with land surface characteristics (e.g., land use, vegetation). However, this study assumes the parameter w is static. Therefore, the trends of ET caused by land surface characteristics are neglected.

**Response:** Thank you for your comments.

We apologize for the confusion. It is true that ω are fitted with annual precipitation, potential evapotranspiration, and actual evapotranspiration with the least-squares regression in this study. Actually, ω in Budyko framework are determined by landscape characteristics (e.g. vegetation cover, soil properties, and topography) (Yang et al., 2008). Generally, ω parameters can be calculated by vegetation changes (Greve et al., 2014). During a long study period (1980-2010), Land surface characteristics significantly affect evapotranspiration by ω parameters. For example, vegetation greening controls interannual evapotranspiration variation (Lu et al., 2021). However, all the four ET data used in this paper assume no interannual vegetation changes (satellite phenology driven), when simulating ET (detailed landcover types in each product have been shown in Table S1). It is worth noting that the assumption mentioned above does not discredit the reliability of the ET products. Furthermore, the contributions of climatic variables to ET trends already include information of the vegetation indirectly. The accuracy of the ET products has been validated in different studies such as Li et al (2018) in China, Wang et al (2018) over the Yellow River basin, and Nooni et al (2019) in the Nile River basin, among others, suggesting good

performances of these products. Therefore, our study only focuses on climatic factors affecting interannual ET changes.

To explain this, we have discussed the influences of vegetation and climate change on ET changes. The text there reads as: such as *"Vegetation can alter water cycle, and energy cycle by biophysical and biochemical feedback to climate change (Forzieri et al., 2020). For example, global surface greening increases ET/transpiration (Lian et al., 2018; Lu et al., 2021), and reduce soil water content (Li et al., 2018a). However, the complex interaction between vegetation and surface makes it difficult to simulate the influence of dynamic vegetation change on ET (Gentine et al., 2019). Meanwhile, strictly disengaging the contributions of climatic variables and vegetation to ET is very difficult due to the interaction between vegetation and climatic variables (Li et al., 2018b). For water-limited regions, precipitation as main water supply to vegetation controls interannual ET changes (Wang et al., 2021). Thus, the dominating factor of interannual ET changes is not vegetation, but rather, atmospheric climate variables (Zhang et al., 2020). Those studies indicate that contribution of climatic variables have already included information of vegetation, indirectly.*

*Given the above reasons, the ET products used in this study do not consider the effect of land use /vegetation changes on ET. When simulating ET, the model frameworks assume no interannual land use changes, so they are regarded as static conditions. Detailed landcover types in each product have been shown in Table S1"*.

Table S1. Comparisons of landcover types data used by the four ET products

| ET product | | Landcover types data | Period |
|---|---|---|---|
| GLEAM3.0a | | MOD44B | Static |
| GLDAS2.0-Noah | | MCD12Q1 | Static |
| MERRA-Land | | Global Land Cover Characterization | Static |
| | W3RA | MOD44B | Static |
| | HTESSEL-CaMa | | Static |
| | JULES | Global Land Cover Characterization | Static |
| EartH2Observe-En | PCR-GLOBWB | | Static |
| | LISFLOOD | GlobCover2009 | Static |
| | HBV-SIMREG | | Static |
| | WaterGAP3 | MOD12Q1 | Static |

Note: However, regarding EartH2Observe-En, the LUC datasets used by seven (in this table) and two models (i.e., ORCHIDEE and SURFEX-TRIP) are available and unavailable, respectively; the LUC is not the necessary input for SWBM.

**References:**

Forzieri, Giovanni; Miralles, Diego G.; Ciais, Philippe; Alkama, Ramdane; Ryu, Youngryel; Duveiller, Gregory; Zhang, Ke; Robertson, Eddy; Kautz, Markus; Martens, Brecht; Jiang, Chongya; Arneth, Almut; Georgievski, Goran; Li, Wei; Ceccherini, Guido; Anthoni, Peter; Lawrence, Peter; Wiltshire, Andy; Pongratz, Julia; Piao, Shilong; Sitch, Stephen; Goll, Daniel S.; Arora, Vivek K.; Lienert, Sebastian; Lombardozzi, Danica; Kato, Etsushi; Nabel, Julia E. M. S.; Tian, Hanqin; Friedlingstein, Pierre; Cescatti, Alessandro. Increased control of vegetation on global terrestrial energy fluxes. Nat. Clim. Chang. 2020, 10, 356–362.

Gentine P, Green J K, Guerin M, et al. Coupling between the terrestrial carbon and water cycles - a review. Environmental Research Letters, 2019, 14(8).

Greve P, Orlowsky B, Mueller B, et al. Global assessment of trends in wetting and drying over land. Nature Geoscience, 2014, 7(10):716-721.

Greve, P., Gudmundsson, L., Orlowsky, B., and Seneviratne, S. I.: A two-parameter Budyko function to represent conditions under which evapotranspiration exceeds precipitation, Hydrol. Earth Syst. Sci., 20, 2195–2205, https://doi.org/10.5194/hess-20-2195-2016, 2016.

Li, S.J., Wang, G.J., Sun, S.L., Chen, H.S., et al., 2018. Assessment of Multi-Source Evapotranspiration Products over China Using Eddy Covariance Observations. Remote Sensing, 210(11), 1692.

Lian, X., Piao, S., Huntingford, C., Li, Y., Zeng, Z., Wang, X., Wang, T., 2018. Partitioning global land evapotranspiration using CMIP5 models constrained by observations. Nature Climate Change 8 (7), 640–646.

Lu, J., Wang, G., Li, S., Feng, A., Zhan, M., Jiang, T., et al. (2021). Projected land evaporation and its response to vegetation greening over China under multiple scenarios in the CMIP6 models. Journal of Geophysical Research: Biogeosciences, 126, e2021JG006327.

Lu, J., Wang, G., Li, S., Feng, A., Zhan, M., Jiang, T., et al. (2021). Projected land evaporation and its response to vegetation greening over China under multiple scenarios in the CMIP6 models. Journal of Geophysical Research: Biogeosciences, 126, e2021JG006327.

Nooni, I.K.; Wang, G.; Hagan, D.F.T.; Lu, J.; Ullah, W.; Li, S. Evapotranspiration and its Components in the Nile River Basin Based on Long-Term Satellite Assimilation Product. Water 2019, 11, 1400.

Wang, G.J., Pan, J., Shen, C.C., Li, S.J., Lu, J., Lou, D., Hagan, D. F. T., et al., 2018. Evaluation of Evapotranspiration Estimates in the Yellow River Basin against the Water Balance Method. Water, 10(12):1884.

Wang, H.N., Lv, X.Z., Zhang, M.Y., 2021. Sensitivity and attribution analysis of vegetation changes on evapotranspiration with the Budyko framework in the Baiyangdian catchment, China. Ecol. Indic. 120, 106963.

Y. Li, S. Piao, L. Z. X. Li, A. Chen, X. Wang, P. Ciais, L. Huang, X. Lian, S. Peng, Z. Zeng, K. Wang, L. Zhou, Divergent hydrological response to large-scale afforestation and vegetation greening in China. Sci. Adv., 2018a, 4, eaar4182.

Yang, H., Yang, D., Lei, Z., & Sun, F. (2008). New analytical derivation of the mean annual water-energy balance equation. Water Resources Research, 44(3), W03410.

Yue Li, Zhenzhong Zeng, Ling Huang, Xu Lian and Shilong Piao. Science. Comment on "Satellites reveal contrasting responses of regional climate to the widespread greening of Earth". Science, 2018b, 360 (6394), eaap7950.

Zhang, D., Liu, X., Zhang, L., Zhang, Q., Gan, R., Li, X., 2020. Attribution of evapotranspiration changes in humid regions of China from 1982 to 2016. J. Geophys. Res.: Atmos., 125 (13), e2020JD032404.

3.  The parameter w is more sensitive to regression in arid climate than in humid climate based on Budyko Equation 1. Therefore, without a detailed study of w, the ET trend analysis in this study may be biased for different climate zones.   In addition, as this study uses four sets of data, it is unclear how w's obtained from each data set are different from each other.

    **Response:** Thank you for your comments.

    We appreciate this suggestion. The fitted parameter ω includes landscape characteristics (e.g. vegetation cover, soil properties, and topography), leading to the parameter with climatic characteristics (Xu et al., 2013). Meanwhile, ET trends of used products in this study are directly related to climate zones. Here, we compare ET trends in each product to climate zones, in which are represented by aridity index. Aridity index (PET/precipitation) in each product is calculated with respective precipitation and PET data. Figure S5(a1-d1) show that the biggest ET trends of all products exit the wettest regions (low aridity index). To study the influence of fitted parameter ω on ET trend analysis, we compare control ET trend (trend$_{CTL}$) to aridity index. The results in Figure S5(a2-d2) show similar results with actual ET trend, meaning the ET trend analysis in the attributed method can capture actual ET change characteristics.

[Figure]

Figure S5. The pixel-wise scatterplots of multi-year average aridity index against actual ET annual values for GLEAM3.0a (a1), EartH2Observe-En (b1), GLDAS2.0-Noah (c1), and MERRA-Land (d1), the control ET trend (trend$_{CTL}$) for GLEAM3.0a (a2), EartH2Observe-En (b2), GLDAS2.0-Noah (c2), and MERRA-Land (d2). Aridity index (PET/precipitation) in each product is calculated with respective precipitation and PET data.

Meanwhile, we also quantify the relationship of parameter ω fitted by precipitation, potential evapotranspiration, and actual evapotranspiration in each product to multi-year average GIMMS NDVI during 1982-2010. Figure 8 shows linear relationship between fitted parameter ω and NDVI for all products with $R^2$ value with 0.13-0.38. In general, parameter ω can be calculated according to the linear relationship between ω and NDVI (Bai et al, 2019; Greve et al, 2014). The results show that our trend analysis keeps the relationship, spatially.

Similar description is also added into the manuscript. The text there reads as: "*Here, we compare ET trends in each product to climate zones, in which are represented by aridity index. Aridity index (PET/precipitation) in each product is calculated with respective precipitation and PET data. Figure S5(a1-d1) show that the biggest ET trends of all products exit the wettest regions (low aridity index). To study the influence of fitted parameter ω on ET trend analysis, we compare control ET trend (trendCTL) to aridity index. The results in Figure S5(a2-d2) show similar results with actual ET trend, meaning the ET trend analysis in the attributed method can capture actual ET change characteristics. Meanwhile, we also quantify the relationship of parameter ω fitted by precipitation, potential evapotranspiration, and actual evapotranspiration in each product to multi-year average GIMMS NDVI during 1982-2010. Figure 8 shows the linear relationship between fitted parameter ω and NDVI for all products with R2 values of 0.13-0.38. In general, parameter ω can be calculated according to the linear relationship between ω and NDVI (Bai et al, 2019; Greve et al, 2014). The results show that our trend analysis keeps the relationship, spatially. We admit that time-varying ω (e.g. vegetation, soil property) will directly affect ET (Lu et al., 2021). The impact of ω would vary as a function of the chosen timescale which requires a more indepth study beyond the scope of the current study*".

[Figure]

Figure 8. Pixel-wise scatterplots of (x-axis) multi-year average NDVI against (y-axis) their fitted ω values in each product. Small letters (a-d) represent GLEAM3.0a, EartH2Observe-En, GLDAS2.0-Noah, and MERRA-Land. GIMMS NDVI data during 1982-2010 is used here.

**References:**

Bai P, Liu X, Zhang D, Liu C. Estimation of the Budyko model parameter for small basins in China. Hydrological Processes. 2019;1–14.

Greve P, Orlowsky B, Mueller B, et al. Global assessment of trends in wetting and drying over land. Nature Geoscience, 2014, 7(10):716-721.

Xu, X., Liu, W., Scanlon, B. R., Zhang, L., & Pan, M.: Local and global factors controlling water-energy balances within the Budyko framework. Geophysical Research Letters, 40, 6123–6129, 10.1002/2013GL058324, 2013.

4. It is a bit confusing on the control experiment setup for sensitivity analysis. The impact of a contributing factor trend on ET trend is analyzed by the difference using 1980 data and the 1980-2010 average (Line 30-34). As there is inter-annual variability in the climate foricngs, why comparing the 1980-year data to 1980-2010 average would reflect the true trend. For example, if a pixel has a decreasing trend in P during 1980-2010 and a dry year in 1980 (i.e., P in 1980 is below average), the experiment setup then would predict an opposite increasing P trend. Therefore, I am not sure if choosing a different year (e.g., 1981) would lead to different results on the trend analysis.

**Response:** Thank you for your comments.

In this study, the sim_CTL experiment can obtain the control ET changes for each product by using all the factors of 1980-2010, and the ET change controlled by one certain factor is simulated by the sensitivity experiment with the factor only in the 1980 and the others factors of 1980-2010. The difference in ET trends between control experiment and each sensitivity experiment is considered as the contribution of that particular climatic variable to ET trends. Actually, choosing a different year in the sensitivity experiment of one factor may lead to different results. In general, there

are two choices (i.e. one year or multi-year average) for this. The two choices are both applied to the attribution analysis of reference evapotranspiration and meteorological drought (Sun et al., 2017; Sun et al., 2019). We compare the PET/precipitation values between 1980s and multi-year average among those products (Figure S1). Overall results show a slight difference between 1980s and multi-year average for PET/precipitation. Similar description is added. The text there reads as: "*The multiyear average can also replace a factor in 1980 during 1980-2010. Figure S1 shows that precipitation and PET values between 1980 and the multiyear average are very close*".

[Figure]

Figure S1. The pixel-wise scatterplots of PET in 1980s against multi-year average PET for GLEAM3.0a (a1), EartH2Observe-En (b1), GLDAS2.0-Noah (c1), and MERRA-Land (d1) and precipitation in 1980s against multi-year average precipitation for GLEAM3.0a (a2), EartH2Observe-En (b2), GLDAS2.0-Noah (c2), and MERRA-Land (d2).

**References:**

Sun, Shanlei; Li, Qingqing; Li, Jinjian; Wang, Guojie; Zhou, Shujia; Chai, Rongfan; Hua, Wenjian; Deng, Peng; Wang, Jie; Lou, Weiping (2019). Revisiting the evolution of the 2009–2011 meteorological drought over Southwest China. Journal of Hydrology, 568, 385–402.

Sun, Shanlei; Chen, Haishan; Sun, Ge; Ju, Weimin; Wang, Guojie; Li, Xing; Yan, Guixia; Gao, Chujie; Huang, Jin; Zhang, Fangmin; Zhu, Siguang; Hua, Wenjian (2017). Attributing the Changes in Reference Evapotranspiration in Southwestern China Using a New Separation Method. Journal of Hydrometeorology, 18(3), 777–798.

---

## Author Response (AR2)

Editor:

Both reviewers were satisfied with the revision. But before acceptance, there are still some presentations problems need to be revised.

**Response:** Thank you very much for your consideration. We are grateful to the reviewer for the very helpful comments.

Reviewer 1:

This study assessed global ET trend through four ET datasets, and quantified the contributions of climatic variables to ET changes through the Budyko control experiment. The results clarify that precipitation, radiation and VPD are the main drivers of evapotranspiration variability and regional characteristics were also generalized. The article has reliable data, reasonable analysis, and detailed uncertainty analysis, but some presentation problems need to be revised.

**Response:** We thank this reviewer for some presentation suggestions, which significantly improves the quality of the manuscript. The following is our detailed responses to the reviewer's comments.

1. The quality of the introduction needs to be improved. The authors need to strengthen the structural logic and avoid the presentation of a list of previous studies, instead using a critical review.

**Response:** Thank you for your comment.

We have already read the introduction, carefully. We have rewritten the research significance part of introduction, such as "*Along these lines, Li et al (2021) attempted to quantify the contribution of those forcing variables to ET trends over China with the Budyko theory. However, there are still unclear questions about the global land ET mechanism. For example, how differently would the conclusions of dominating ET factors over water-limited regions be for global dry lands? Which variable controls ET over the global tropical zone is unclear, despite the results of VPD controlling ET over the energy-limited region of China. Which variable controls ET over the boreal region is unclear. For example, Precipitation, air temperature, and radiation control Amazon's ET changes (Pan et al., 2020), while significantly increased ET in the humid region mostly results from increasing air temperature (Wang et al., 2022). For boreal region, increasing air temperature is significantly correlated with ET (Wang et al., 2022), while increasing VPD contributes to ET process (Helbig et al., 2020). Therefore, it is necessary to assess global ET mechanisms using the same attribution method for solving these problems*" in Line 28-37 of page 2.

2. P2 Line10, "Those results have indicated that the improper choice of ET models and forcing data may add significant uncertainties to the ET attributions.": "improper" is probably not the right word.

**Response:** Thank you for your comment.

The word "improper" has been changed to the word "inappropriate". Please see line 14 in page 2.

3. P2 Line 15, "Studies have indicated that increased VPD primarily determines the recent PET increase, a function of air temperature and humidity (Dai et al., 2017; Ficklin et al., 2017). Increased VPD tends to make plants close their stomata to avoid water loss and thus restrain transpiration (Novick et al., 2016; McAdam et al., 2015).": It does not read well and lacks connecting words.

**Response:** Thank you for your comment.

We have added a connecting word "however" in Line 17-20 of page 2, such as "*Studies have indicated that increased VPD primarily determines the recent PET increase, which is a function of air temperature and humidity (Dai et al., 2017; Ficklin et al., 2017), however increased VPD tends to make plants close their stomata to avoid water loss and thus restrain transpiration (Novick et al., 2016; McAdam et al., 2015)*".

4. P2 Line 20, "Li et al. (2021) have found VPD has dominated the increase of annual ET in energy-limited regions such as southeastern China.": This sentence lacks connection with the previous one.

**Response:** Thank you for your comment.

We have added a connecting sentence in Line 21-22 of page 2, such as "Therefore, it is very important to clarify how VPD affects long-term ET changes".

5. P12 Line 20, "Those studies indicate that contribution of climatic variables have(has) already included information of vegetation, indirectly." may be misleading. The authors describe **the ET database as static in terms of land use, indicating that this study does not consider the contribution of land use change, as stated in the next paragraph**. If the authors are trying to show that land use has a limited impact, they can refer to this article (Hoek van Dijke A J, Herold M, Mallick K, et al. Shifts in regional water availability due to global tree restoration[J]. Nature Geoscience, 2022, 15(5): 363-368.).

**Response:** Thank you for your comment and sharing paper.

We have already read this paper, carefully. And, we have changed the sentence in Line 23-24 of page 12 to "Those studies indicate that vegetation influences on ET already contain the signal of climatic variables, which are essential for vegetation growth".